# Identification of Novel Genes and Proteoforms in *Angiostrongylus costaricensis* through a Proteogenomic Approach

**DOI:** 10.3390/pathogens11111273

**Published:** 2022-10-31

**Authors:** Esdras Matheus Gomes da Silva, Karina Mastropasqua Rebello, Young-Jun Choi, Vitor Gregorio, Alexandre Rossi Paschoal, Makedonka Mitreva, James H. McKerrow, Ana Gisele da Costa Neves-Ferreira, Fabio Passetti

**Affiliations:** 1Instituto Carlos Chagas, Fiocruz, Curitiba 81350-010, PR, Brazil; 2Laboratory of Toxinology, Oswaldo Cruz Institute, Fiocruz, Rio de Janeiro 21040-900, RJ, Brazil; 3Laboratory of Integrated Studies in Protozoology, Oswaldo Cruz Institute, Fiocruz, Rio de Janeiro 21040-360, RJ, Brazil; 4Department of Medicine, Washington University School of Medicine, St. Louis, MO 63110, USA; 5Bioinformatics and Pattern Recognition Group (Bioinfo-CP), Department of Computer Science (DACOM), Federal University of Technology-Parana (UTFPR), Cornélio Procópio 86300-000, PR, Brazil; 6Center for Discovery and Innovation in Parasitic Diseases, Skaggs School of Pharmacy and Pharmaceutical Sciences, University of California, San Diego, CA 92093, USA

**Keywords:** RNA-Seq, mass spectrometry, nematode, genome annotation, ncRNAs

## Abstract

RNA sequencing (RNA-Seq) and mass-spectrometry-based proteomics data are often integrated in proteogenomic studies to assist in the prediction of eukaryote genome features, such as genes, splicing, single-nucleotide (SNVs), and single-amino-acid variants (SAAVs). Most genomes of parasite nematodes are draft versions that lack transcript- and protein-level information and whose gene annotations rely only on computational predictions. *Angiostrongylus costaricensis* is a roundworm species that causes an intestinal inflammatory disease, known as abdominal angiostrongyliasis (AA). Currently, there is no drug available that acts directly on this parasite, mostly due to the sparse understanding of its molecular characteristics. The available genome of *A. costaricensis*, specific to the Costa Rica strain, is a draft version that is not supported by transcript- or protein-level evidence. This study used RNA-Seq and MS/MS data to perform an in-depth annotation of the *A. costaricensis* genome. Our prediction improved the reference annotation with (a) novel coding and non-coding genes; (b) pieces of evidence of alternative splicing generating new proteoforms; and (c) a list of SNVs between the Brazilian (Crissiumal) and the Costa Rica strain. To the best of our knowledge, this is the first time that a multi-omics approach has been used to improve the genome annotation of *A. costaricensis*. We hope this improved genome annotation can assist in the future development of drugs, kits, and vaccines to treat, diagnose, and prevent AA caused by either the Brazil strain (Crissiumal) or the Costa Rica strain.

## 1. Introduction

One of the critical steps after genome sequencing and assembling is annotating the genomic features [1]. Many computational tools perform ab initio genome annotation using only genome sequence motifs [2,3,4,5,6,7]. However, predicting genes and splice variants within a genome sequence using this approach is particularly challenging for eukaryotic genomes because genes are usually distant from each other and are interrupted by introns [8]. Therefore, data from multiple high-throughput technologies such as RNA sequencing (RNA-Seq) and protein mass spectrometry (MS/MS) are often used to assist in predicting these genomic features [9]. Usually, RNA-Seq reads are aligned to a reference genome, followed by gene prediction and the assembly of transcript sequences [10,11]. Transcript sequences are computationally translated, generating a customized protein-sequence database for protein identification by MS/MS [12,13]. Integrating RNA-Seq and MS/MS data improves genome annotation because the protein-coding genes’ expression and distinct splice variants can be confirmed at the transcript and protein levels [14,15,16].

In addition to gene annotation and alternative splicing prediction, genetic variations, such as single-nucleotide variants (SNVs) and insertions and deletions (INDELs), can also be detected during genome annotation using RNA-Seq and MS/MS data [17]. SNVs are the most prevalent genetic variation in eukaryotes and are generated by point mutations in the genome sequence of a given species [18]. These SNVs may change a codon in protein-coding genes and eventually induce a single-amino-acid substitution (SAAV), a missense mutation [19]. SNVs can be detected by identifying mismatches in the alignment of RNA-Seq reads onto the genome sequence [20]. SNVs of missense mutations can also be confirmed at the protein level by recognizing their correspondent SAAVs in the MS/MS data [21]. Identifying SNVs and SAAVs is particularly important in nematodes because point mutations have been associated with resistance to anthelmintic drugs [22].

The genome sequences of pathogenic helminths are, in many cases, draft versions in need of annotation improvement [23]. *Angiostrongylus costaricensis* is a nematode species that causes an intestinal inflammatory disease known as abdominal angiostrongyliasis (AA) [24]. AA was first reported in 1952 [25], and the parasite was described in 1971 in Costa Rica [26]. This disease is a public health problem in Latin America, especially in Costa Rica and Brazil [27]. Currently, no drugs acting directly on this parasite are available, primarily due to the sparse understanding of its molecular characteristics [28]. The current publicly available genome draft of *A. costaricensis* is specific to the Costa Rica strain, and its annotation is not supported by transcript- or protein-level evidence. Although the Brazilian strain (Crissiumal strain, Rio Grande do Sul, Brazil) is well-characterized in terms of its morphological aspects [29], migratory pathways [30], and vascular pathology [31], a characterization from a genetic perspective is still needed. Therefore, this study aimed to improve the genome annotation of *A. costaricensis* based on RNA-Seq and MS/MS data with novel coding and non-coding genes and alternative splicing proteoforms. We also provide a list of SNVs between the Brazilian (Crissiumal) and the Costa Rica strain. To the best of our knowledge, this is the first time a multi-omics approach has been used to assist in the genome annotation of *A. costaricensis*.

## 2. Materials and Methods

### 2.1. Source of Animals and Worm Isolation

*Sigmodon hispidus* (from the family Cricetidae) were bought from Vyrion Systems (USA) in 1991 and used to establish a colony that has been maintained to present in the Laboratory of Pathology, Oswaldo Cruz Institute, Fiocruz, Rio de Janeiro, Brazil (license IBAMA 34095). Animals were kept in a temperature-controlled room (approximately 21/22 °C) and fed ad libitium. All experimental procedures were performed in accordance with the ethical recommendations of the Animal Ethics Committee of the Oswaldo Cruz Foundation (CEUA/Fiocruz) (licenses LW 43/13 and LW 26/15). The calculation of statistical power was not applicable to this study. Instead, the number of *S. hispidus* necessary for the isolation of worms in sufficient quantity to perform the RNA-Seq in triplicate and MS/MS in quadruplicate was used. Adult *A. costaricensis* (Crissiumal strain) were recovered from the mesenteric arteries of twenty-five *S. hispidus* 30–40 days post-infection. Rodent euthanasia was performed with an intraperitoneal injection of sodium thiopental at a dose of 120 mg/kg body weight. The life cycle of the parasites was maintained at the laboratory through their successive passages in mollusks *Biomphalaria glabrata* (intermediate hosts) and rodents *S. hispidus* (definitive hosts), as previously described [30]. Animal procedures were approved by the Ethics Committee on the Use of Animals at the Oswaldo Cruz Institute (CEUA L-039/2017). Worms were washed at least three times in PBS (pH 7.4) to remove host tissues and were frozen in liquid nitrogen. Frozen worms were ground to a fine powder using pre-chilled mortar and pestle on dry ice.

### 2.2. RNA Sequencing

Pulverized frozen tissue (~100–200 mg) was placed into microtubes containing 1 mL of TRIzol (Life Technologies, Carlsbad, CA, USA). DNA was degraded using the PureLink™ DNase Set according to the manufacturer’s instruction (Life Technologies). Total RNA was isolated using the PureLink RNA Mini-Kit (Ambion, Austin, TX, USA) according to the manufacturer’s instruction (Life Technologies) and quantified with a Nanodrop ND-1000 spectrophotometer (Thermo Scientific, Waltham, MA, USA, EUA). The RNA integrity was assessed via agarose gel electrophoresis and via automated electrophoresis (2100 Bioanalyser Agilent, Santa Clara, CA, USA), wherein only samples with RNA integrity number (RIN) equal or greater than 8 were selected. All research activities carried out with the Brazilian genetic heritage (law 13123/15 and Decree 8772/16) were registered in the National System of Genetic Resource Management and Associated Traditional Knowledge (SisGen register A26AB0E). Sequencing libraries were prepared from total RNA for all six samples (three adult female, three adult male) using the Illumina TruSeq Stranded Total RNA kit [32]. Paired-end sequence data were generated on the Illumina HiSeq 2500 platform targeting 10 gigabases per sample. Analytical processing of data started using unaligned bam files.

### 2.3. Mass Spectrometry

Protein extracts were obtained after mechanical maceration (Sample Grinding kit, GE Healthcare, Chicago, IL, USA) of males and females (4 mg each) in 100 µL of 0.05 M Tris-HCl pH 7.6, containing 2% SDS. After maceration, samples were incubated in the extraction buffer for 10 min at room temperature, boiled in a water bath for 5 min, and centrifuged. The supernatants were collected, and their protein content was estimated by the bicinchoninic acid (BCA) method [33], using bovine serum albumin (BSA) as standard protein. Aliquots of each protein extract were mixed (*v*/*v*) with 0.05 M Tris-HCl pH 7.6, containing 2% SDS and 100 mM DTT, followed by boiling for 5 min and incubation at room temperature for an additional 15 min. Then, 100 µg of the reduced extracts were subjected to the FASP (Filter Aided Sample Preparation) protocol [34] using Microcon-30kDa ultrafiltration membranes (Merck-Millipore, Burlington, MA, USA, Cat. No. MRCF0R030). Following conventional peptide alkylation and washing steps, two enzymes were sequentially used for protein digestion (Lys-C and trypsin, 12–18 h at 37 °C each enzyme, E:S 1:100 *w*/*w*). For each biological replicate, a single final digest was desalted in POROS R2 microcolumns and processed for nLC-nESI-MS/MS. All analyses were done in the Orbitrap QExactive Plus mass spectrometer (Thermo Scientific), hyphenated to the Dionex Ultimate 3000 nanochromatograph, as previously described [35], unless stated otherwise. Fractionation was performed on a New Objective PicoFrit column (tip 10 μm, 75 μm inner diameter × 40 cm length) packed in-house with Reprosil Pur 120 C18-AQ 1.9 μm resin (Dr. Maisch GmbH, Ammerbuch, Germany). The following mobile phases were used: (A) 0.1% formic acid in water; (B) 0.1% formic acid in acetonitrile. The column was eluted with a linear gradient of 2–45%B for 162 min, followed by a gradient between 45–80%B for 4 min, and, finally, isocratic washing with 80%B for 2 min. Each biological replicate (n = 8) was acquired in technical triplicate.

### 2.4. Protein Identification and Quantification

Protein identification was based on the peptide–spectral matching (PSM) approach, using the Comet search algorithm, and implemented in the freely available PatternLab for Proteomics computational environment (version 5) [36]. A single FASTA file containing non-redundant protein sequences from the WormBase database (9344 entries), supplemented with in silico translated transcripts identified by the software BRAKER (12,228 entries) and with protein sequences containing SAAVs predicted by the software Pilon (16,574 entries), was used by the search engine. The “Generate Search DB” module generated a target-reverse database enriched with common MS contaminant sequences (e.g., keratins, albumin, and trypsin). Uninterpreted high-resolution MS/MS spectra were searched against this comprehensive database using Comet default parameters. Enzyme specificity was semi-specific, no proline restriction was specified for trypsin, up to 2 missed cleavages were allowed, and the initial precursor mass tolerance was set to 40 ppm. The following modifications were considered (up to 2 variable modifications per peptide): (1) Carbamidomethyl (C, fixed); (2) Carbamidomethyl (DEHK (including N-terminal), variable); (3) Deamidation (NQ, variable). PSM results were filtered by the search engine processor (SEPro) tool implemented in PatternLab V. The final post-processing step was adjusted to converge to reliable results, showing ≤1% FDR at spectra, peptide, and protein levels and ≤10 ppm mass error for MS1 and MS2 spectra.

The same identification strategy was applied for proteome quantification, although using a database containing all the above target sequences, except for those containing the SAAVs predicted by the software PILON. Protein label-free quantification was performed according to the normalized ion abundance factor (NIAF) [36]. A minimum of seven MS1 points were accepted for obtaining the extracted ion-chromatogram (XIC) of unique peptides only, and at least two unique peptides were required for quantifying each protein.

### 2.5. Annotation of Novel Gene and Splicing Variants Using RNA-Seq Data

RNA-Seq raw reads were trimmed according to the base call quality (PHRED score > 20) and the presence of sequence adaptors, using the software Trim Galore version 0.4.0 [37]. Trimmed reads were aligned to the publicly available draft genome sequence of *A. costaricensis* (assembly version A_costaricensis_Costa_Rica_0011_upd, WormBase WBPS15), using the software HISAT2 version 2.1.0 [38]. Reads uniquely aligned to the genome were selected. The alignment file in SAM format was sorted by read names and converted to its binary format, BAM, using the SAMtools version 1.3.1 package [39]. The BAM files of each biological replicate were merged and sorted by genome coordinates using the SAMtools package. The merged BAM file was used as extrinsic evidence for the genome annotation software, BRAKER version 2.0 [40], generating the final genome annotation file of protein-coding genes in GTF format (general transfer format). The genome annotation of *A. costaricensis* (annotation version 2014-06-50HGPpatch, WormBase version WBPS15) was improved with the BRAKER annotation. This process also merged gene models with overlapped transcripts according to BRAKER and WormBase annotations to reduce artifactual fragmentation. The transcript sequences and *in silico* translations of complete ORFs were obtained using GffRead version 0.12.1 (http://ccb.jhu.edu/software/stringtie/gff.shtml, accessed on 21 August 2022).

### 2.6. Functional Annotation

Blast2GO version 5.2.5 [41] and InterProScan from the package OmicsBox version 1.3.11 [42] were applied to predict the functional annotations of complete ORFs from the improved WBPS15 annotation file. Parameters used were *in silico* translated sequences of complete ORFs (21,584 entries) as queries and Metazoa sequences from nr v5 database as the reference sequence for Blastp search. The subsequent functional annotation steps were performed using Blast2GO default parameters.

### 2.7. Annotation of Single-Nucleotide Variants (SNVs)

The publicly available draft genome of *A. costaricensis* (WormBase version WBPS15) is from the Costa Rica strain. However, the RNA-Seq dataset generated in this work is from the Brazil strain (Crissiumal). Therefore, a genome sequence with SNVs identified in the Brazil strain (Crissiumal) and not mandatorily in the Costa Rica strain was built using the alignments of the RNA-Seq reads to the reference genome and the Pilon version 1.24 software [43] with default parameters. The ANNOVAR software (version 2020-06-07) [44] was used to identify whether the identified SNVs would cause amino acid substitutions. A Perl script was used to verify whether the substitutions were between amino acids with similar physico-chemical characteristics (conservative) or not (non-conservative). Complete ORFs in the genome sequence with Brazilian SNVs were translated *in silico* using the improved WBPS15 GTF file and the GffRead software [45] to build the protein-sequence database used in protein identification.

### 2.8. Annotation of Non-Coding RNA Genes

The non-coding RNA (ncRNAs) genes were predicted using structure strategy and sequence-similarity search. For the former, we used the following software: (i) INFERNAL version 1.1.4 (9 December 2020) [46] with cmsearch software, parameter-*cut_ga* (to decrease the false-positive results) and based on covariance models from Rfam database version 14.7 (December 2021, 4069 families) [47]. For the latter, we used BLASTN [48] against RNAcentral release 19 [49] with parameters dust, *e-value 0.00001,* and filtering by identity and query coverage with at least 95% as the threshold. We built custom Python scripts for filtering and merging results. When the prediction from both approaches overlapped, the one with the minor e-value score was chosen. Finally, we compared our prediction to the WormBase, and those ncRNAs that did not coincide with any WormBase ncRNA were selected as the set of novel ncRNAs.

### 2.9. Annotation of Single-Amino-Acid Variants (SAAVs)

The identified peptides containing SAAVs were compared to the protein sequences using the aligner blastp (version 2.9.0) [48]. For the sake of confidence, beyond the previously described FDR threshold of 1%, the analysis considered only peptides identified with primary scores ≥ 2.5 (for charge states +2) or ≥3.0 (for charge states ≥ +3). The alignment results were used to determine the correct coordinates of each peptide within its protein sequence. Using an in-house Perl script, the exact amino acid substitution within the identified peptide was obtained using the ANNOVAR output file, with the coordinate of each amino acid substitution within each protein sequence.

### 2.10. Transcript Quantification

Counting information of read fragments per transcript was obtained using the HTSeq-count software (version 3.3.2) [50]. The DESeq2 package (version 1.32.0) [51] from the R Bioconductor toolset imported the counting information to a data frame. The GenomicFeatures package (version 1.44.2) [52] from the R Bioconductor toolset was used to import genomic features from the genome annotation file (GTF file) and to normalize the counting data using the Fragments Per Kilobase Million (FPKM) method.

### 2.11. Analysis of Transcriptome and Proteome Abundance Levels

FPKM and NIAF values were used to infer the abundance of the transcripts and proteins, respectively. These estimates were used to check the correspondence between transcriptome and proteome abundances. To make FPKM and NIAF values comparable, they were log_10_ transformed, and z-scores were calculated using the *scale* native function of the R program language. For a visual comprehension of expression patterns, transcripts and proteins with similar abundances were clustered using the k-means clustering algorithm (k = 15) and plotted as a heatmap using the R package ComplexHeatmap (version 3.13) [53].

## 3. Results

### 3.1. Improving the A. costaricensis Genome Annotation

Our approach used RNA-Seq data to search for novel genes and transcripts, including protein-coding and non-coding genes and alternative transcripts. The resulting annotation files (GFF3) are available at https://github.com/Matheusdras/Acostaricensis-genome-reannotation.

#### 3.1.1. Novel Protein-Coding Genes and Transcript Variants

On average, 71% of the RNA-Seq-trimmed reads were uniquely mapped onto the *A. costaricensis* genome sequence, and 52% were assigned to genes using the current WormBase reference genome annotation (Appendix A). The RNA-Seq read alignments were used as extrinsic evidence for gene prediction using the BRAKER software [40]. The reference annotation was then improved with the genome annotation derived from RNA-Seq read alignments (BRAKER). Gene models with overlapped transcripts according to BRAKER and WormBase annotations were merged on the final improved annotation to reduce artifactual fragmentation (example in Appendix A). After combining gene models, the number of genes from BRAKER annotation decreased from 13,136 to 11,531, and WormBase gene annotations were reduced from 13,417 to 12,229.

The improved annotation encompasses 14,588 genes, 27,788 mRNAs, and 21,584 complete ORFs (Table 1), including the prediction of 2359 novel genes, 2553 novel transcripts, and 10,194 novel transcript variants (Figure 1 and examples in Appendix A). On average, each novel gene had one transcript (Appendix A, yellow) with five exons (Appendix A, yellow), while novel transcript variants had twelve exons (Appendix A, light green). The mean length of the novel transcripts was 552 base pairs (bp) (Appendix A, yellow) and of the novel transcript variants was 1427 bp (Appendix A, orange). The mean length of exons from the novel transcripts was 98 bp (Appendix A, yellow) and from novel transcript variants was 115 bp (Appendix A, orange). The length of each transcript and its category can be found in Appendix A.

#### 3.1.2. Novel Non-Coding Genes

The subsequent analysis was focused on genomic regions with no non-coding gene annotations with support of RNA-Seq reads. A set of 426 non-coding RNAs (ncRNAs) from different classes were computationally annotated, including transfer RNA (tRNA), Piwi-interacting RNA (piRNA), ribosomal RNA (rRNA), small RNA (sRNA), and small nucleolar RNAs (snoRNA). Among the novel ncRNAs annotated, 30 had the support of RNA-Seq reads with FPKM values ˃ 1 (Figure 2, Appendix A).

#### 3.1.3. Functional Annotation

Next, we used computational predictions to provide a function annotation to our improved version of the *A. costaricensis* genome. Overall, 72% of the 21,584 complete ORF sequences from the BRAKER-improved version of the Wormbase genome could be annotated by Blast2GO [41] (Table 2). Among the annotated protein sequences, 7273 were derived from WormBase and 7672 from BRAKER. Most of the annotated protein sequences are related to metabolic processes and nucleic acid binding and are assigned as integral membrane components (Appendix A and Appendix A). Most of the protein sequences were inferred from electronic annotation (IEA) (Appendix A) based on the species *Angiostrongylus cantonensis* (Appendix A). Most of the enzymes annotated are hydrolyzes (Appendix A). The protein domain that was more abundant in the annotation was protein kinase (Appendix A).

### 3.2. Protein Identification Using a Customized Protein-Sequence Database

In addition to identifying novel genes and transcript variants, RNA-Seq reads were used to call SNVs based on the identification of homozygous or heterozygous genomic positions using the PILON software [43]. The resulting variant calling file (VCF) is available at https://github.com/Matheusdras/Acostaricensis-genome-reannotation. In total, 554,066 SNVs were computationally predicted, including 552,181 homozygous (allele frequency 100%) and 18,855 heterozygous (median allele frequency 98%). A set of 410,270 SNVs occurred in protein-coding genes, resulting in 262,441 synonymous, 147,025 missenses, 612 stop codon gains, and 192 stop codon losses (Appendix A).

A customized non-redundant protein-sequence database containing the WormBase database improved with in silico translated transcripts by the BRAKER software and protein sequences with SAAVs predicted by the PILON software was built. This customized protein-sequence database was used to assess the impact of the predicted proteins/proteoforms and missense polymorphisms in the proteome using MS/MS data. When confirmed on the proteome data (Figure 3a,b), such SNVs were appointed as SAAVs.

As depicted in Figure 4a, the identification of novel protein-coding genes and proteoforms (BRAKER software) allowed for the identification of a higher number of peptides in the MS/MS data, followed by the WormBase database and the strategy to identify SAAVs (PILON software). These peptides were mapped to 4296 protein sequences determined using the PILON software, 4085 sequences obtained using the BRAKER strategy, and 2587 protein sequences belonging to the original WormBase database. Protein sequences in the improved database used in the MS/MS spectra search were not redundant. Therefore, protein sequences that originated from the different approaches did not show intersections (Figure 4b). The BRAKER software allowed for the identification of a higher number of unique peptides as compared to the WormBase database or the PILON software (Figure 4c). Regarding the number of unique proteins identified, the BRAKER software showed a better performance (51% of all unique identifications), followed by the PILON software (33%) and the WormBase database (16%) (Figure 4d).

Figure 5a shows the mean number of SNVs observed for each transcript and the impact on the amino acid translation prediction. The majority of SNVs resulted in synonymous amino acid substitutions. The conservation of amino acid substitutions was assessed based on the physico-chemical properties of the residues. Those cases with missense polymorphisms were computationally translated, generating 70,674 conserved and 76,351 non-conserved potential SAAVs. On average, 1.64 SAAVs were identified per peptide in the case of conservative substitutions, whereas 1.75 SAAVs were detected for non-conservative ones (Figure 5b). In total, 1419 peptides with conserved SAAVs and 1488 peptides with non-conserved SAAVs were identified (Appendix A).

### 3.3. Transcriptome and Proteome Quantification

A total of 27,788 mRNAs were quantified across six biological replicates, with a 56.66 FPKM average normalized abundance level (Appendix A and Appendix A). Using the identification results from the Wormbase + BRAKER database, 2600 proteins were quantified across 8 biological replicates, with a mean NIAF value of 7 × 10^−4^ (Appendix A and Appendix A). When comparing mRNA and protein abundances, 1612 pairs were grouped in 15 clusters according to their normalized abundance levels, revealing concordant and discordant clusters (Figure 6 and Appendix A).

## 4. Discussion

Multi-omics data have been largely applied to assist the annotation of human [54,55] and mouse [56] genomes, and RNA-Seq data has also been used for that purpose in previous nematode studies [57,58]. Recently, Logan and colleagues (2020) used RNA-Seq and MSMS data to improve the genome annotation of *Necator americanus*, being the first multi-omics analysis of a parasitic nematode [9]. In our analysis, we have identified novel genes and exons, with support of RNA-Seq read alignments, that are not found in the reference genome annotation available in the WormBase database [23]. These results motivated us to propose an improved version of this genome annotation built on RNA-Seq and MS/MS data.

Our prediction using the software BRAKER revealed 2359 novel hypothetical genes and 10,194 novel hypothetical transcript variants, most of them (99%) supported by transcriptome evidence and 80% with FPKM values ˃ 1. The MS/MS data analysis using the customized protein-sequence database confirmed the transcriptome findings at the protein level, revealing that most identified peptides are exclusive to the novel predicted proteins.

Interestingly, the WormBase database’s entries allowed the identification of only 16% of the unique proteins described in this study. The remaining 84% of the proteins identified with proteotypic peptides were distributed as follows: (a) A total of 51% were identified as a result of our BRAKER-based strategy to find novel protein-coding genes and proteoforms; and (b) 33% were identified based on protein sequences containing high-confidence SAAVs. The functional annotation of the complete ORFs was also incorporated in the improved version of the genome annotation. In addition to revealing novel genes and gene products, this approach also provided expression evidence from mRNA and protein levels for known genes. It is important to draw attention to the fact that we assessed the transcriptome and proteome of adult worms, and the gene expression of other life-cycle stages should be further evaluated in future studies.

Geographic separation leads to genetic diversity between populations due to mutations accumulated within nematode genomes over a long period of time [59]. This study used RNA-Seq and MS/MS data to better elucidate DNA sequence polymorphisms between the Brazilian (Crissiumal) and the Costa Rica strains, revealing SNVs and SAAVs at both transcript and protein levels, respectively. These results corroborate a previous study in which sequences from the mitochondrial *cox1* gene were analyzed, revealing significant nucleotide differences in *A. costaricensis* between the Brazil and Costa Rica isolates [60]. Additionally, other similar results were found between the mitochondrial genomes of these two strains [61]. Thus, strains from different geographical locations might represent cryptic or separate species and should be studied further [62]. We believe that these results can be used in future studies to investigate the evolutive history of the Costa Rica and Brazil strains and ultimately unveil new possibilities for treating the disease in both countries. We expect our results to encourage other groups to use a similar strategy when studying distinct strains.

The annotation of non-coding RNA genes is often incipient or unavailable in nematodes with sequenced genomes, such as *A. costaricensis*, except for the model organism *Caenorhabditis elegans*. Therefore, we also performed a computational prediction of ncRNAs to complement the annotation of the genome of *A. costaricensis*, predicting 426 ncRNA genes that were not included in the current WormBase genome annotation. Although mRNAs were enriched during the RNA-Seq library construction through poly(A)+ selection, some non-coding RNAs were still detected in our data, especially rRNAs. This is expected because rRNAs usually represent more than 80% of a given transcriptome [63], and poly(A)+ selection protocols are not 100% efficient [64]. Thus, even though the annotation of ncRNA genes was not the primary aim of this study, we could benefit from the small fraction of contaminant ncRNAs present in our data to provide experimental evidence for the prediction of ncRNAs.

We used clustering analysis applied to RNA-Seq and MS/MS data as an attempt to study the transcriptome/proteome regulation. As a result, two clusters with proportional mRNA and protein levels and three clusters with inversely proportional mRNA and protein levels were detected. These results indicate that some mRNAs may be subjected to regulatory elements preventing their translation, and others may have higher translation rates, as reviewed by Kumar and colleagues [65]. However, these results should be carefully interpreted because the correlation between transcript and protein levels can be affected by technical limitations [55], and the actual correspondence between transcriptomes and proteomes is still a topic of debate in the literature [66]. However, we believe that our results provide some insights into the mechanisms of the post-transcriptional and post-translational regulations of *A. costaricensis*.

As reported by other studies, the use of multi-omics data as source of genetic information is an efficient approach for genome annotation [67,68,69]. Here, we propose a pipeline for the genome annotation of helminthics: (1) Align RNA-Seq reads onto the reference genome sequence (Hisat2 software); (2) use RNA-Seq read alignment as extrinsic evidence for gene/transcript prediction (BRAKER software); (3) use RNA-Seq read alignment as extrinsic evidence for SNV calling (Pilon software); (4) select transcripts with SNVs that cause putative amino acid substitutions; (5) perform a computational translation of the complete ORFs to build a customized protein-sequence database containing putative SAAVs (GffRead software); (6) perform a protein database search on MS proteomic data using the customized protein-sequence database (PatternLab for proteomics software); (7) analyze the identified peptides and proteins to confirm at proteome level those genes and SNVs predicted at the transcriptome level. All codes and command-line parameters used in our analysis are publicly available at https://github.com/Matheusdras/Acostaricensis-genome-reannotation.

## 5. Conclusions

In summary, we applied RNA-Seq and MS/MS data to refine the genome annotation of *A. costaricensis*. This improved genome annotation encompasses novel protein-coding and non-coding genes, transcript variants/proteoforms, and a list of SNVs and SAAVs. We hope these results can advance the knowledge of *A. costaricensis* protein-coding and non-coding genes and motivate future researchers to drive new hypotheses regarding this parasite, mainly focusing on abdominal angiostrongyliasis treatment. Additionally, in many cases, multi-omics data are publicly available and can be reused and integrated [70]. Therefore, we believe that the use of integrative omics of nematodes causing neglected diseases is a promising strategy to unveil potential targets for anti-helminthic drugs, vaccine development, and diagnoses.

## Figures and Tables

**Figure 1 pathogens-11-01273-f001:**
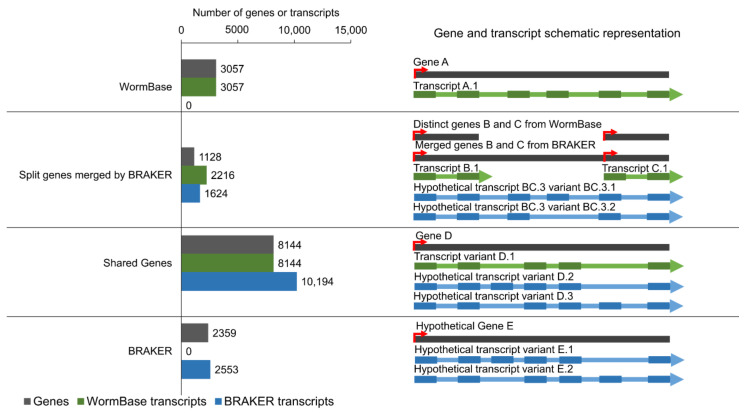
Distribution of different gene and transcript categories in the improved genome annotation. Left bar plots show the number of genes (grey) and transcripts predicted by the WormBase (green) or the BRAKER (blue) annotations. The schematic representations of different categories of genes and transcripts are shown on the right part of the figure. Rectangles represent exons, lines indicate introns, and arrows show the orientation of the genes/transcripts. Red arrows represent transcription start sites. **WormBase**: reference genes and transcripts which were only annotated by the WormBase; **Split genes merged by BRAKER**: fragmented genes in the WormBase which were merged by BRAKER prediction and their respective transcripts; **Shared Genes**: genes predicted both by WormBase and BRAKER genome annotations and their respective transcripts; **BRAKER**: novel hypothetical genes and transcripts predicted only by BRAKER genome annotation (Appendix A).

**Figure 2 pathogens-11-01273-f002:**
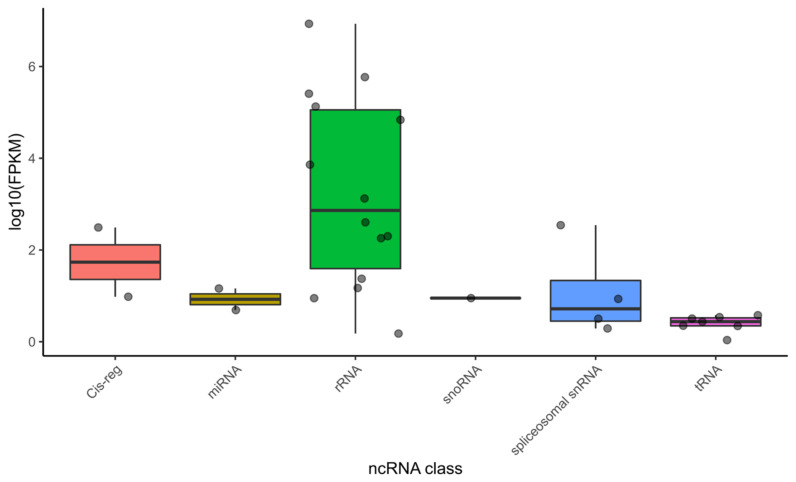
Distribution of ncRNAs abundance levels in *A. costaricensis* RNASeq dataset. Dots represent ncRNAs clustered by class along the *x*-axis. The normalized abundance value (FPKM) of each ncRNA is represented on the log_10_ scale on the *y*-axis.

**Figure 3 pathogens-11-01273-f003:**
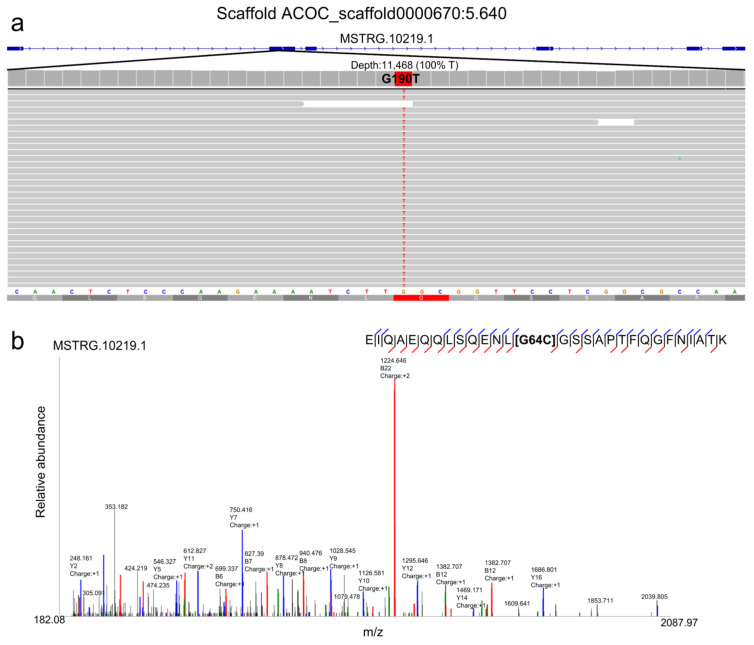
Polymorphism identified in gene product MSTRG.10219.1. (**a**) IGV screenshot showing G190T SNV from MSTRG.10219.1 transcript; (**b**) annotated tandem mass spectrum of a unique peptide from protein MSTRG.10219.1 showing G64C SAAV, which corresponds to G190T SNV.

**Figure 4 pathogens-11-01273-f004:**
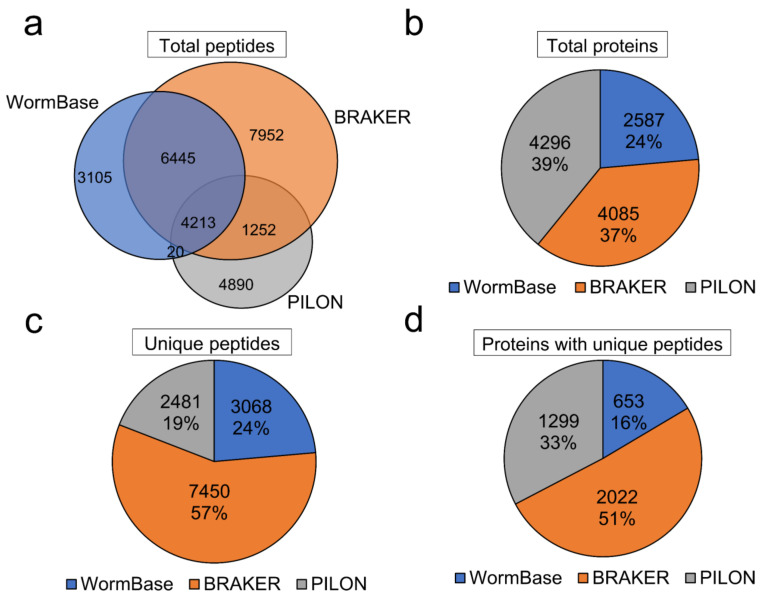
Comparison of the number of peptide and protein identifications observed by employing different strategies to determine the fasta sequences that make up the protein database. (**a**) Venn diagram showing the relationship among the identified peptides (total) using WormBase, BRAKER, and PILON; pie charts showing the number of total proteins inferred (**b**), unique peptides identified (**c**), and proteins inferred with unique peptides (**d**) using each strategy.

**Figure 5 pathogens-11-01273-f005:**
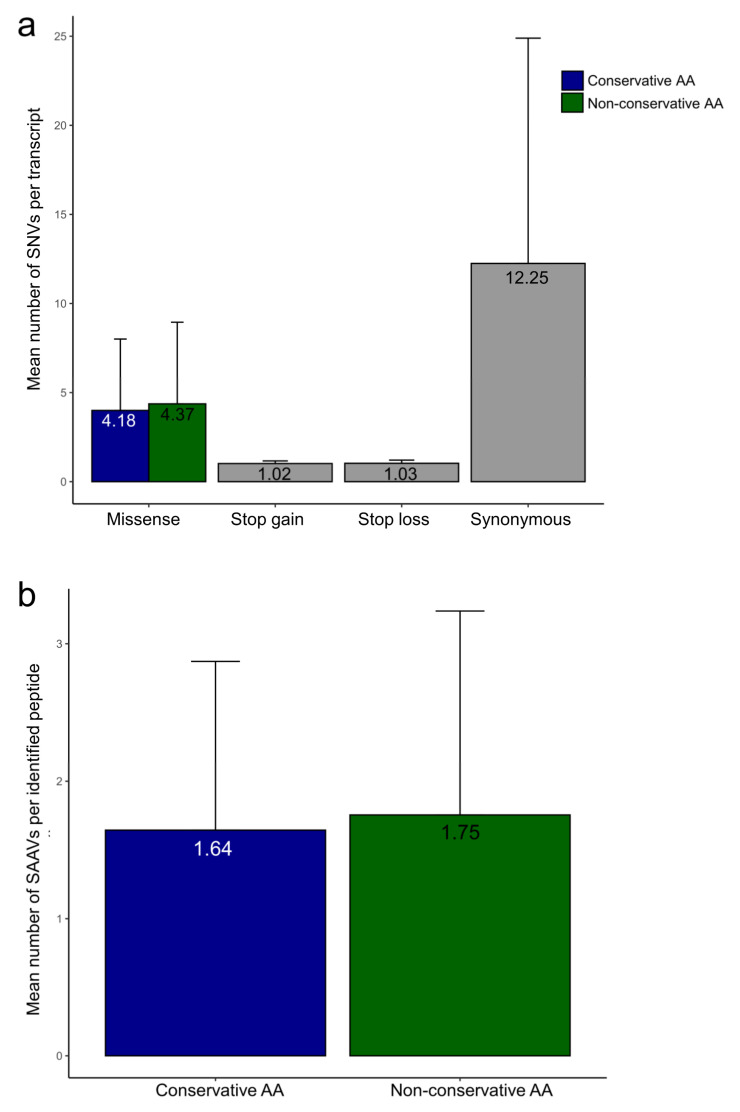
Distribution of SNV and SAAVs in *A. costaricensis* RNASeq and proteomic datasets. (**a**) The bar plot shows the mean number of SNVs per transcript from non-synonymous, stop codon gain, stop codon loss, and synonymous polymorphisms; (**b**) the bar plot shows the mean number of SAAVs per identified peptide from conservative and non-conservative amino acid (AA) substitutions.

**Figure 6 pathogens-11-01273-f006:**
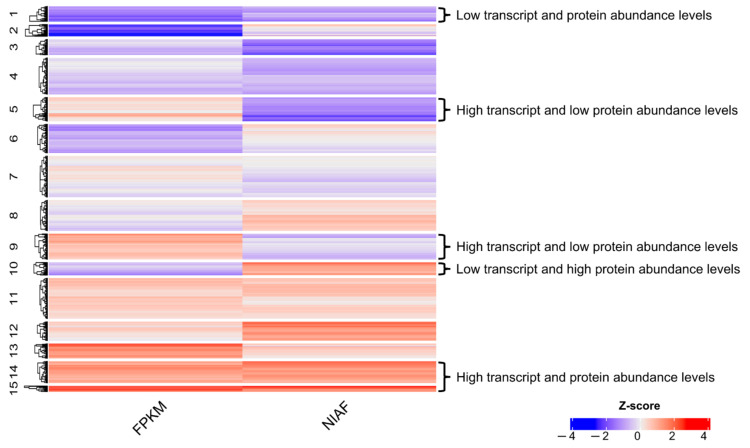
Heatmap showing 15 clusters of mRNAs and proteins. Clusters are based on the z-score of mRNAs and proteins normalized abundance levels (FPKM and NIAF). Blue rectangles represent mRNAs and proteins with low FPKM and NIAF values, respectively. Red rectangles represent mRNAs and proteins with high FPKM and NIAF values, respectively. Clusters with clear concordant abundance levels (clusters 1 and 15) and discordant abundance levels (clusters 5, 9, and 10) are highlighted by braces and proper description.

**Table 1 pathogens-11-01273-t001:** The number of genes, mRNAs, and complete ORFs of WormBase, BRAKER, and WormBase improved annotations.

Annotation Source	Genes	mRNAs	Complete ORFs
WormBase	13,417	13,411	12,154
BRAKER	13,136	15,630	13,914
WormBase improved with BRAKER’s annotation	14,588	27,788	21,584

**Table 2 pathogens-11-01273-t002:** Distribution of functional annotation of complete ORFs.

Functional Annotation Status	Number of Sequences	Fraction of Sequences
Blast hits	20,945	97%
Interpro hits	18,036	84%
Blast hits and mapped GO terms	17,343	80%
Complete Blast2GO annotation	15,612	72%
Interpro hits and mapped GO terms	10,847	50%
No InterPro hits	3548	16%
No blast hits	639	3%

## Data Availability

All codes used to generate the results can be found on GitHub via URL https://github.com/Matheusdras/Acostaricensis-genome-reannotation. All RNA sequencing data in this study are available at the NCBI (National Center for Biotechnology Information) SRA (Sequence Read Archive) repository under accession number PRJNA851259. This study’s MS/MS proteomics data are available at the PRIDE (PRoteomics IDEntifications database) repository via ProteomeXchange Consortium under the accession number PXD034605.

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
