# Peer review of "Identification of Novel Genes and Proteoforms in Angiostrongylus costaricensis through a Proteogenomic Approach"

_pathogens, 2022, doi:10.3390/pathogens11111273_

Round 1

Reviewer 1 Report

The article provides a substantial step in the knowledge about the A. costaricencis genome, that is relevant for future functional studies, in particular.

There are relatively minor issues that should be addressed, though. The answers to questions should be included in the manuscript, particularly if explanations are asked.

Generic issues:

1.       The Discussion is full of references to figures, tables and supplementary files. The Discussion should summarise the results and discussing them, without the need for the reader to refer back to figures or tables. In particular

2.       When referring to variants, it should always be clear that it is in relation to the Costa Rican strain of A. costaricencis, the only available public genome draft. As such, the authors should also not use the adjective “specific”, as not enough genomes have been surveyed. It is better to refer to polymorphisms between the two genomes, or similar.

3.       In the Materials and Methods, it would help the reader if the Mass spectrometry section (2.3) as placed immediately before section 2.8 (Protein identification and quantitation).

4.       The parameters for the in house Perl scripts should be specified, so that the procedures could be replicated by others.

5.       The authors could provide a summary recommended pipeline for other researchers who want to conduct a similar multi-omics genome annotation.

Specific comments:

1.       Line 57: “are generated by” rather than “consist of”

2.       Lines 65-66: rephrase. Odd sentence. What “molecular aspects”? The knowledge about the genome, or genetics, may be incipient, rather. Also, the genomes are not draft, although the genome sequences may be draft.

3.       Introduction (line 79): some information should be given about the Brazilian strain (Crissiumal), with an explanation as to why it was chosen for this work.

4.       Line 91: specify here the strain.

5.       Line 118: was the method of worm collection different from the method of collection for RNA sequencing?

6.       Lines 170-172: odd phrasing. With only RNA-Seq data it is not possible to assemble a full genome sequence, let alone with only specific SNVs. The authors have aligned the RNA-Seq reads to the reference genome to identify SNVs between the two genomes. Any assembled data would not be a full genome sequence, but a “transcribed genome sequence”. And if only SNVs were used, it would not be possible to determine if they would cause amino acid substitutions. Please explain.

7.       Line 416: geographical separation leads to genetic diversity between populations, with time, because mutations accumulate in each population and the two populations are kept separate. However, geographical separation by itself does not introduce genetic diversity between (within does not make sense) genomes.

8.       Lines 417-419: This information should have been in the Introduction.

9.       Lines 423-427: the authors mention the cox1 gene in relation to the genome data, and then mention the results between mitochondrial genomes. It should be noted that cox1 is a mitochondrial gene, and it would make more sense to be mentioned in relation to the mitochondrial genomes.

Author Response

Response to Reviewer 1 (R1) Comments

We would like to thank Reviewer 1 to his/her comments.

R1-Q1. The Discussion is full of references to figures, tables and supplementary files. The Discussion should summarise the results and discussing them, without the need for the reader to refer back to figures or tables. In particular

Response to R1-Q1: We appreciate the reviewer’s comment and we have removed these references from the discussion as suggested.

R1-Q2. When referring to variants, it should always be clear that it is in relation to the Costa Rican strain of A. costaricencis, the only available public genome draft. As such, the authors should also not use the adjective “specific”, as not enough genomes have been surveyed. It is better to refer to polymorphisms between the two genomes, or similar.

Response to R1-Q2: We have changed the term “specific” to SNVs between the Brazilian (Crissiumal) and the Costa Rica strain or SNVs identified in the Brazil strain (Crissiumal) and not in the Costa Rica strain as suggested.

R1-Q3. In the Materials and Methods, it would help the reader if the Mass spectrometry section (2.3) as placed immediately before section 2.8 (Protein identification and quantitation).

Response to R1-Q3: We appreciate the reviewer’s comment and we have altered the order of these sections as suggested.

R1-Q4. The parameters for the in house Perl scripts should be specified, so that the procedures could be replicated by others.

Response to R1-Q4: We appreciate the reviewer’s comment. To make our analysis replicable and easy to access in a more comprehensible way we have made publicly available all codes and command-line’s parameters used in this work at the repository: https://github.com/Matheusdras/Acostaricensis-genome-reannotation. This information can be found at the Data Availability Statement of the article.

R1-Q5. The authors could provide a summary recommended pipeline for other researchers who want to conduct a similar multi-omics genome annotation.

Response to R1-Q5: We appreciate the reviewer’s comment. We added a new paragraph (lines 457-470) to provide a summary recommended pipeline as suggested.

Specific comments:

R1-Q6. Line 57: “are generated by” rather than “consist of”

Response to R1-Q6: We appreciate the reviewer’s comment and we have changed the sentence as suggested.

R1-Q7. Lines 65-66: rephrase. Odd sentence. What “molecular aspects”? The knowledge about the genome, or genetics, may be incipient, rather. Also, the genomes are not draft, although the genome sequences may be draft.

Response to R1-Q7: We appreciate the reviewer’s comment. This sentence has been changed to: “The genome sequences of pathogenic helminths are, in many cases, draft versions in need of annotation improvement”.

R1-Q8. Introduction (line 79): some information should be given about the Brazilian strain (Crissiumal), with an explanation as to why it was chosen for this work.

Response to R1-Q8: We appreciate the reviewer’s comment. We added a new sentence (lines 73-76) providing the proper explanation for this matter.

R1-Q9. Line 91: specify here the strain.

Response to R1-Q9: We appreciate the reviewer’s comment and we have added the strain used to produce our data (Crissiumal strain).

R1-Q10. Line 118: was the method of worm collection different from the method of collection for RNA sequencing?

Response to R1-Q10: We appreciate the reviewer’s comment. In fact, the worm collection method used in both MS and RNA-Seq were the same, we have deleted the sentence “A. costaricensis adult worms were collected as previously described” at line 118 to avoid misleading interpretation.

R1-Q11. Lines 170-172: odd phrasing. With only RNA-Seq data it is not possible to assemble a full genome sequence, let alone with only specific SNVs. The authors have aligned the RNA-Seq reads to the reference genome to identify SNVs between the two genomes. Any assembled data would not be a full genome sequence, but a “transcribed genome sequence”. And if only SNVs were used, it would not be possible to determine if they would cause amino acid substitutions. Please explain.

Response to R1-Q11: We appreciate the reviewer’s comment. As described in the section 2.5 (line 169), RNA-Seq read alignments were used as extrinsic evidence for gene/transcript prediction, accomplished using the software BRAKER. Furthermore, as described in the section 2.7 (line 194), RNA-Seq read alignments were also used to identify SNVs between the Brazil and Costa Rica strains and generate a genome sequence representing the Brazil strain’s SNVs. The putative role on amino acid substitution was determined by the software ANNOVAR, as described in the 2.9 section (line 219), and some of the SAAVs were experimentally confirmed in MS data as described in the 2.9 and 3.2 (line 219 and 317) sections. We hope we could have properly addressed the reviewer’s comment.

R1-Q12. Line 416: geographical separation leads to genetic diversity between populations, with time, because mutations accumulate in each population and the two populations are kept separate. However, geographical separation by itself does not introduce genetic diversity between (within does not make sense) genomes.

Response to R1-Q12: We appreciate the reviewer’s comment. We have rephrased the sentence to: Geographic separation leads to genetic diversity between populations due to mutations accumulated within nematode genomes over a long period of time.

R1-Q13. Lines 417-419: This information should have been in the Introduction.

Response to R1-Q13: We appreciate the reviewer’s comment. We have relocated the sentence to the introduction (lines 73-76): “Although the Brazilian strain (Crissiumal strain, Rio Grande do Sul, Brazil) is well-characterized in terms of its morphological aspects, migratory pathways, and vascular pathology a characterization from a genetic perspective is still needed.”

R1-Q14. Lines 423-427: the authors mention the cox1 gene in relation to the genome data, and then mention the results between mitochondrial genomes. It should be noted that cox1 is a mitochondrial gene, and it would make more sense to be mentioned in relation to the mitochondrial genomes.

Response to R1-Q14: We agree with the reviewer’s comment and the proper corrections were made (lines 424-429).

Reviewer 2 Report

The manuscript by Gomes Da Silva describes the comprehensive study using a proteogenomic approach to improve the gene annotations within the Angiostrongylus costaricensis genome. The methods are well described and the results displayed well. As the authors have discussed, this is a much needed approach for several parasitic helminths to improve genome assemblies/gene annotations. I have a few minor comments that should be addressed prior to the manuscript being suitable for publication.

1. The introduction reiterates the abstract in part, often using the exact wording/sentence structure. Could the authors add further background information to the introduction to broaden the scope discussed.

2. The authors used adult parasites for their transcriptome and proteome analysis in this study, despite maintaining the life cycle where they had access to the other life cycle stages that would have enhanced this study. Similar, the authors need to further discuss this point in their discussion that they may be missing stage-specific genes/proteins in their analysis as they didn't incorporate other life cycle stages.

3. Line 175 - physical or physico? -  as used in the discussion

4. Line 395 - being the first

Author Response

Response to Reviewer 2 Comments

We would like to thank Reviewer 2 to his/her comments.

R2-Q1. The introduction reiterates the abstract in part, often using the exact wording/sentence structure. Could the authors add further background information to the introduction to broaden the scope discussed.

Response to R2-Q1: We agree with the reviewer’s comment. We used this opportunity to provide background information regarding the Brazilian strain (lines 73-76) and refine the introduction writing (lines 76-79).

R2-Q2. The authors used adult parasites for their transcriptome and proteome analysis in this study, despite maintaining the life cycle where they had access to the other life cycle stages that would have enhanced this study. Similar, the authors need to further discuss this point in their discussion that they may be missing stage-specific genes/proteins in their analysis as they did not incorporate other life cycle stages.

Response to R2-Q2: We appreciate the reviewer’s comment. The first stage larvae (L1) is found in rodent stool and the infective third stage larvae (L3) is found in intermediate hosts (mollusks Biomphalaria glabrata). However, adult worms are the ones found in human mesenteric arteries and for this reason are the most important phase in the life cycle of this parasite on the medical perspective [1]. Therefore, in our study we analyzed the transcriptome and proteome of adult worms. We have properly addressed this subject in the discussion section (lines 417-419) as suggested.

R2-Q3. Line 175 - physical or physico? - as used in the discussion

Response to R2-Q3: We appreciate the reviewer’s comment. The correct form is physico-chemical, the proper correction was made accordingly (line 203 and line 364).

R2-Q4. Line 395 - being the first

Response to R2-Q4: We appreciate the reviewer’s comment. The proper correction was made accordingly (line 397).

References

  1. Rojas, A.; Maldonado-Junior, A.; Mora, J.; Morassutti, A.; Rodriguez, R.; Solano-Barquero, A.; Tijerino, A.; Vargas, M.; Graeff-Teixeira, C. Abdominal angiostrongyliasis in the Americas: fifty years since the discovery of a new metastrongylid species, Angiostrongylus costaricensis. Parasit. Vectors 2021, 14, 374, doi:10.1186/s13071-021-04875-3.